# Deep CO$_2$ in the end-Triassic Central Atlantic Magmatic Province

Manfredo Capriolo [1✉], Andrea Marzoli [1], László E. Aradi [2], Sara Callegaro [1,3], Jacopo Dal Corso [4], Robert J. Newton[4], Benjamin J.W. Mills [4], Paul B. Wignall[4], Omar Bartoli [1], Don R. Baker [5], Nasrrddine Youbi [6,7,8], Laurent Remusat[9], Richard Spiess [1] & Csaba Szabó[2]

Large Igneous Province eruptions coincide with many major Phanerozoic mass extinctions, suggesting a cause-effect relationship where volcanic degassing triggers global climatic changes. In order to fully understand this relationship, it is necessary to constrain the quantity and type of degassed magmatic volatiles, and to determine the depth of their source and the timing of eruption. Here we present direct evidence of abundant CO$_2$ in basaltic rocks from the end-Triassic Central Atlantic Magmatic Province (CAMP), through investigation of gas exsolution bubbles preserved by melt inclusions. Our results indicate abundance of CO$_2$ and a mantle and/or lower-middle crustal origin for at least part of the degassed carbon. The presence of deep carbon is a key control on the emplacement mode of CAMP magmas, favouring rapid eruption pulses (a few centuries each). Our estimates suggest that the amount of CO$_2$ that each CAMP magmatic pulse injected into the end-Triassic atmosphere is comparable to the amount of anthropogenic emissions projected for the 21$^{st}$ century. Such large volumes of volcanic CO$_2$ likely contributed to end-Triassic global warming and ocean acidification.

[1] Department of Geosciences, University of Padova, 35131 Padova, Italy. [2] Lithosphere Fluid Research Lab, Institute of Geography and Earth Sciences, Eötvös Loránd University, Budapest H-1117, Hungary. [3] Centre for Earth Evolution and Dynamics, University of Oslo, 0371 Oslo, Norway. [4] School of Earth and Environment, University of Leeds, Leeds LS2 9JT, UK. [5] Department of Earth and Planetary Sciences, McGill University, Montreal H3A 0E8, Canada. [6] Department of Geology, Faculty of Sciences-Semlalia, Cadi Ayyad University, Marrakesh, Morocco. [7] Instituto Dom Luiz, University of Lisbon, 1749-016 Lisbon, Portugal. [8] Faculty of Geology and Geography, Tomsk State University, Tomsk 634050, Russia. [9] Muséum National d'Histoire Naturelle, Sorbonne University, UMR CNRS 7590, Institut de Minéralogie, de Physique des Matériaux et de Cosmochimie, 75005 Paris, France. ✉email: manfredo.capriolo@phd.unipd.it

Volatile elements affect the behaviour of magmas during their rise through the crust, and control the timing and energy of volcanic eruptions. When rapidly released into the atmosphere, volcanic gases such as CO, $CO_2$, $CH_4$, $SO_2$, $H_2S$, HCl, and $CH_3Cl$, can have a devastating impact on the global climate and biota[1–3]. The best example from the geologic record is the emplacement of large igneous provinces (LIPs)[4], which are synchronous with several major Phanerozoic mass extinctions, indicating LIPs as potential triggers of global-scale climatic and environmental changes via the release of volatiles[2]. LIPs, often volumetrically dominated by continental flood basalts, are exceptional intraplate magmatic events involving huge magma volumes (up to $10^6$ km$^3$)[5], that are emplaced episodically, leading to a pulsed release of their volatile phases[6,7]. This potentially results in rapid rise of atmospheric $CO_2$ and global climate warming[8–10].

In addition to perturbing the climate, volcanic $CO_2$ plays a key role in the storage, ascent and eruption of magma, and drives the stability and evolution of magma reservoirs, regulating flood basalt magmatism and associated degassing fluxes[11,12]. The importance of exsolved volatile phases (e.g., $CO_2$-rich fluids) is highlighted by recently developed models of magmatic plumbing systems[13,14]. In these new models, magma reservoirs are dominated by a crystalline mush, forming a multi-phase (i.e., solid, liquid and gas) system in which crystals, melt, and exsolved volatiles can interact and ascend independently towards the surface. Due to the low solubility of $CO_2$ in silicate melts, exsolution of $CO_2$-rich fluids from melts occurs deep in the crust (i.e., within magmatic plumbing system) or even in the upper mantle[15], whereas at shallow depths the fluids typically become more $H_2O$-rich[16]. Since the exsolution of $CO_2$ changes the physical properties (e.g., density, viscosity and buoyancy) of magmas, it may therefore play a crucial role in their ascent, and could explain the pulsed eruptive style observed for LIPs. However, direct evidence of $CO_2$ abundance in the deep magmas of LIPs is lacking.

In this study, we investigate the history of volatiles in the magmas of the Central Atlantic Magmatic Province (CAMP), one of Earth's largest LIPs[17,18], by analysing volatiles in melt inclusions (MIs), particularly $CO_2$. We examine the implications of these findings for magma eruption history and subsequent impact on the global climate. The emplacement of CAMP (peak activity at 201.6–201.1 Ma) occurred during the early stages of the Pangaea supercontinent break-up, leading to the opening of the Central Atlantic Ocean, and is synchronous with the End-Triassic Extinction (ETE), one of the five most severe biotic crises during the Phanerozoic[19–22]. At least $3 \times 10^6$ km$^3$ of CAMP basaltic magmas were erupted or intruded into the continental crust over an area of $10^7$ km$^2$ in brief pulses, from a few centuries to a few millennia each, characterized by high eruption rates[6,23]. Such short and powerful eruptions may have had a severe impact on global climate by limiting the time in which negative feedback processes, such as the weathering of Ca–Mg silicates, can abate warming and acidification. CAMP magmatism coincided in time with three marked negative carbon isotope excursions bracketing the main extinction period[20,24,25], and with an inferred strong rise of atmospheric $CO_2$[8,9]. In general, the pulsed magmatic and degassing activities of LIPs[6,7] can cause a rapid rise of atmospheric $CO_2$ and greenhouse conditions, which are reflected by rapid δ$^{13}$C negative excursions recorded in both organic matter and carbonates[10], testifying to a global perturbation of the exogenic (i.e., superficial) carbon cycle. A rapid input of $^{13}$C-depleted volatile phases into the atmosphere–hydrosphere system is possibly triggered by the emission of volcanic $CO_2$[24], and likely enhanced by the emission of $CO_2$ and $CH_4$ derived from the thermal metamorphism of intruded organic matter-rich sediments[26].

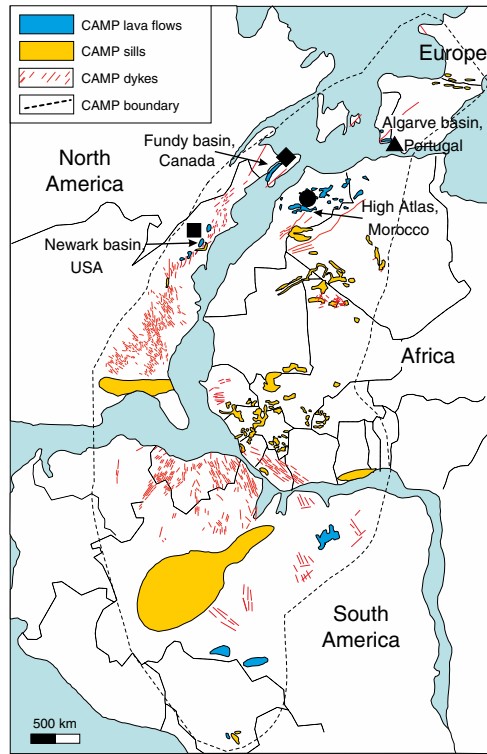

**Fig. 1 Map of CAMP in central Pangea at about 200 Ma.** The black symbols indicate the provenance of the studied samples: triangle for Portugal, circle for Morocco, square for New Jersey, USA, and diamond for Nova Scotia, Canada. The figure is modified after ref. [23].

Here, we screened a suite of over 200 intrusive and effusive samples from CAMP basaltic lava flows and sills in North America (USA and Canada), Africa (Morocco) and Europe (Portugal; Fig. 1 and Supplementary Table 1), and combined several in situ analytical techniques to investigate the presence of $CO_2$ within MI bubbles and constrain their formation depth. Our multidisciplinary analytical approach reveals that gas exsolution bubbles trapped in MIs are a previously unappreciated direct proxy of volatile species degassed during LIP magmatic activity[27–29]. In the case of CAMP, our analysis confirms the abundance of $CO_2$ (up to $10^5$ Gt volcanic $CO_2$ degassed during CAMP emplacement) and indicates that at least part of this carbon has a middle- to lower-crust or mantle origin, suggesting that CAMP eruptions were rapid and potentially catastrophic for both climate and biosphere.

## Results and discussion
About 10% of the >200 investigated intrusive and effusive CAMP basaltic rocks show gas exsolution bubble-bearing MIs, hosted mainly in clinopyroxene and occasionally in plagioclase, orthopyroxene and olivine (Supplementary Figs. 1 and 2). The studied CAMP basaltic rocks are mainly porphyritic and microcrystalline, and the principal mineral phases are labradoritic-bytownitic plagioclase, augitic (abundant) and pigeonitic (scarce) clinopyroxene, rare Mg-rich orthopyroxene, and rare and mostly altered Mg-rich olivine. As accessory mineral phases, magnetite is common, while ilmenite is rare. In effusive rock samples (from USA, Canada, Morocco and Portugal), glomerocrystic aggregates of augitic clinopyroxene and plagioclase are commonly present (Supplementary Fig. 1), and are interpreted as clots of partially crystallized mineral mush from the transcrustal magmatic plumbing system[13,14]. In the only studied intrusive sample (from

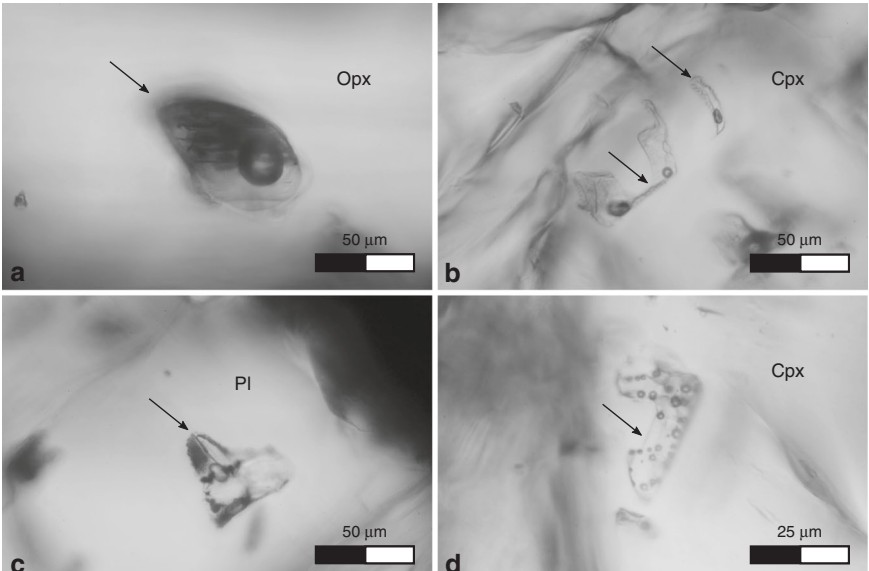

**Fig. 2 Representative bubble-bearing melt inclusions at transmitted light optical microscopy.** The black arrows indicate the bubble-bearing melt inclusions. **a** Single-bubble MI hosted in orthopyroxene (Opx; sample NS21, Nova Scotia, Canada). **b** Single- and multi-bubble MIs, with very irregular shapes, hosted in augitic clinopyroxene (Cpx; sample AN137A, Morocco). **c** Multi-bubble MI, partially crystallized (containing also opaque mineral phases), hosted in calcic plagioclase (Pl; sample NS9, Nova Scotia, Canada). **d** Multi-bubble MI hosted in augitic clinopyroxene (Cpx; sample NS9, Nova Scotia, Canada).

Palisades sill, USA), olivine is abundant and usually well preserved (Supplementary Fig. 2).

MIs are nearly ubiquitous in glomerocrystic aggregates of clinopyroxene and plagioclase (Supplementary Fig. 1). The bubble-bearing MIs usually have irregular shapes, can be single- or multi-bubble MIs, and contain up to 25 bubbles per inclusion, displaying a large range of glass/bubble ratios even within the same host crystal or crystal clot (i.e., there is no proportionality between the volume of glass and the volume/number of bubbles; Fig. 2 and Supplementary Fig. 2). In detail, the estimated volume fraction of bubbles within each MI ranges from <0.1 to >0.5 approximately. Moreover, MIs present a great variability in size, approximately from 5 to 50 μm on the principal axis. Bubbles within them usually have spherical shape and generally range from 1 to 15 μm in diameter (Supplementary Fig. 2). Sometimes bubbles are aggregated in the MIs, probably due to post-entrapment coalescence (Supplementary Fig. 2). Some MIs are partially crystallized, containing μm-sized daughter minerals in addition to, or instead of, bubbles. These crystals, likely formed from the melt after the entrapment, are mainly opaque mineral phases, such as sulphides and oxides (e.g., magnetite). The MIs glass has a more silicic (mainly andesitic) and more differentiated composition compared to the host basaltic rocks, and is clearly different from typical CAMP basalts or basaltic andesites (Supplementary Fig. 3 and Supplementary Table 2). The MIs glass is generally enriched in $SiO_2$ and $Al_2O_3$, and depleted in FeO, MgO and CaO compared to the host rocks (Supplementary Fig. 4), and would correspond to a residual melt after fractionation of ca. 40% augitic clinopyroxene, 10% plagioclase and 5% magnetite from a typical CAMP basalt (see "Methods" section). Such differentiation can only partly be due to post-entrapment crystallization of the few tiny crystals within the MIs or of the host clinopyroxene[30–32], which displays constant augitic composition, shows only faint chemical zonation towards the glass, and is substantially out of equilibrium with it (see "Methods" section). The most evident compositional zoning of the host clinopyroxene consists of a decrease in CaO content and a slight increase in both MgO and FeO content close to the contact with the MIs (Fig. 3 and

Supplementary Fig. 5). Hence, the local thin rim around the MIs of slightly Ca-depleted and Fe ± Mg-enriched clinopyroxene suggests the probable presence of augite–pigeonite exsolution lamellae close to the boundary of MIs, which likely formed at subsolidus conditions from an intermediate composition clinopyroxene that crystallized from the entrapped melt (i.e., post-entrapment crystallization). However, the chemical disequilibrium between the MIs glass and the host clinopyroxene, and the lack of significant chemical zoning within the host clinopyroxene at the contact with MIs are not consistent with substantial diffusive re-equilibration within the host clinopyroxene and suggest a rapid cooling after melt entrapment. This indicates that a previously differentiated bubble-bearing melt was entrained between interstices of growing crystals, and rapidly cooled down, forming MIs.

The bubbles within MIs were investigated in all the samples by confocal Raman microspectroscopy (Supplementary Table 3), looking for carbon species (CO, $CO_2$, $CH_4$ and elemental C), as well as for other important volatile compounds in volcanic systems ($SO_2$, $H_2S$ and $H_2O$). In almost all analysed bubbles, $CO_2$ (within 54 bubbles of 9 samples) or elemental carbon (within 41 bubbles of 2 samples) were detected in both single- and multi-bubble MIs of rock samples collected from all over the CAMP (Fig. 4 and Source Data 1 and 2). In detail, Raman spectra show that $CO_2$ in CAMP bubbles is characterized by low density (ca. 0.1 $g/cm^3$; see "Methods" section), and elemental carbon in CAMP bubbles is characterized by low crystallinity (i.e., it is present as disordered graphite and amorphous carbon; see "Methods" section). $CO_2$ concentrations from 0.5 to 1.0 wt% in whole MIs (i.e., glass plus bubbles) were calculated from the density of gaseous $CO_2$ within the bubbles (Supplementary Table 4) and from the estimated volume fraction of these bubbles within MIs (see "Methods" section). Other volatiles such as CO, $CH_4$, $SO_2$ and $H_2S$ were not detected, while $H_2O$ was often found within the glass of MIs (Supplementary Fig. 6), but never in the bubbles. The MIs glass, investigated through Nano-SIMS, contains about 0.5–0.6 wt% $H_2O$ and 30–90 ppm $CO_2$ (Supplementary Table 5).

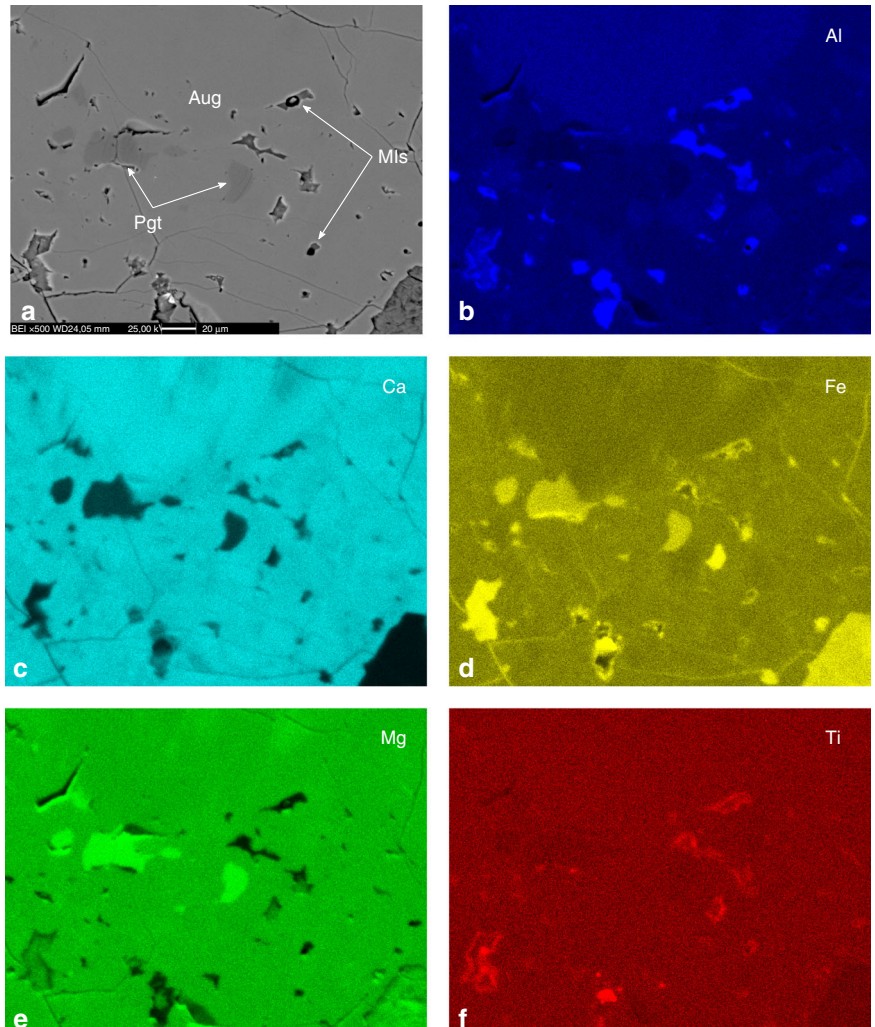

**Fig. 3 Chemical maps of glomerocrystic clinopyroxene aggregates.** Backscattered electrons (BSE) image **a** and corresponding scanning electron microscopy with energy-dispersive X-ray spectroscopy (SEM–EDS) maps **b**–**f** of a thin section area including MIs and the hosting glomerocrystic clinopyroxene aggregates. In the BSE image the brighter portions of clinopyroxene have augitic (Aug) composition and the darker ones have pigeonitic (Pgt) composition. In the SEM–EDS maps the brighter regions correspond to higher concentrations of the analysed element. These maps were acquired on sample NEW31 (New Jersey, USA). The scale bar is shown in **a**. **a** BSE image, **b** Al map, **c** Ca map, **d** Fe map, **e** Mg map, and **f** Ti map.

**$CO_2$ in CAMP basalts**. The analysed bubble-bearing MIs strongly suggest that the CAMP magmatic system was rich in $CO_2$. Most of the analysed bubbles contains $CO_2$ or, less frequently, elemental carbon, and no detectable amounts of any other investigated volatile phase (Supplementary Note 1). In particular, confocal Raman microspectroscopy allowed to distinguish and characterize both $CO_2$ and elemental carbon (see "Methods" section). The Raman spectrum of $CO_2$ is characterized by two sharp bands, usually called Fermi diad or Fermi doublet, associated to two symmetrical weak bands, usually called hot bands[33] (Fig. 5a and Source Data 1). Instead, the first-order Raman spectrum of elemental carbon is characterized by two different bands, the composite D band, activated in disordered graphite by lattice defects and typical of non-crystalline structures[34,35], and the single G band, typical of graphite[34] (Fig. 5b and Source Data 2). This last band is here employed in a crossplot to characterize the different types of elemental carbon, distinguishing disordered graphite and amorphous carbon (Fig. 6). Interestingly, $CO_2$ and elemental carbon within gas exsolution bubbles are never present together in the same samples. The elemental carbon, which is likely present as a thin film coating the inner spherical surface of the bubbles, replaces $CO_2$ in some samples, probably due to a change in the oxidation state within MIs,

for instance related to a diffusive loss of oxygen from the bubbles to the melt, when the latter crystallized oxides during cooling. The large variability in volume and number of bubbles observed in coexisting MIs (ranging from 1 to 25 bubbles per MI, approximately occupying from <0.1 to >0.5 of the MI volume, as optically estimated in thin and thick sections) reveals heterogeneous entrapment of MIs[27,36]. Therefore, the bubbles within MIs are interpreted as gas exsolution bubbles, formed during exsolution of a $CO_2$-rich fluid phase likely from the silicate melt prior to, or during, their entrapment. Gas exsolution within MIs after melt entrapment was probably of minor importance, particularly for MIs with large bubbles[27], because the trapping of a bubble-free melt would have produced homogeneous MIs, displaying very similar glass/bubble ratios, which were not observed in this study. The volatile-saturated melt and the volatiles may have a cogenetic origin (i.e., the melt was entrapped along with volatiles immediately after, or during, gas exsolution), or may have different origins (i.e., the melt was entrapped along with volatiles exsolved from deeper magmas, or degassed and fluxed from intruded crustal rocks).

Clinopyroxene compositions and volatile element concentrations suggest that $CO_2$ entrapment occurred within the deep magmatic roots of CAMP. The pressure of crystallization of

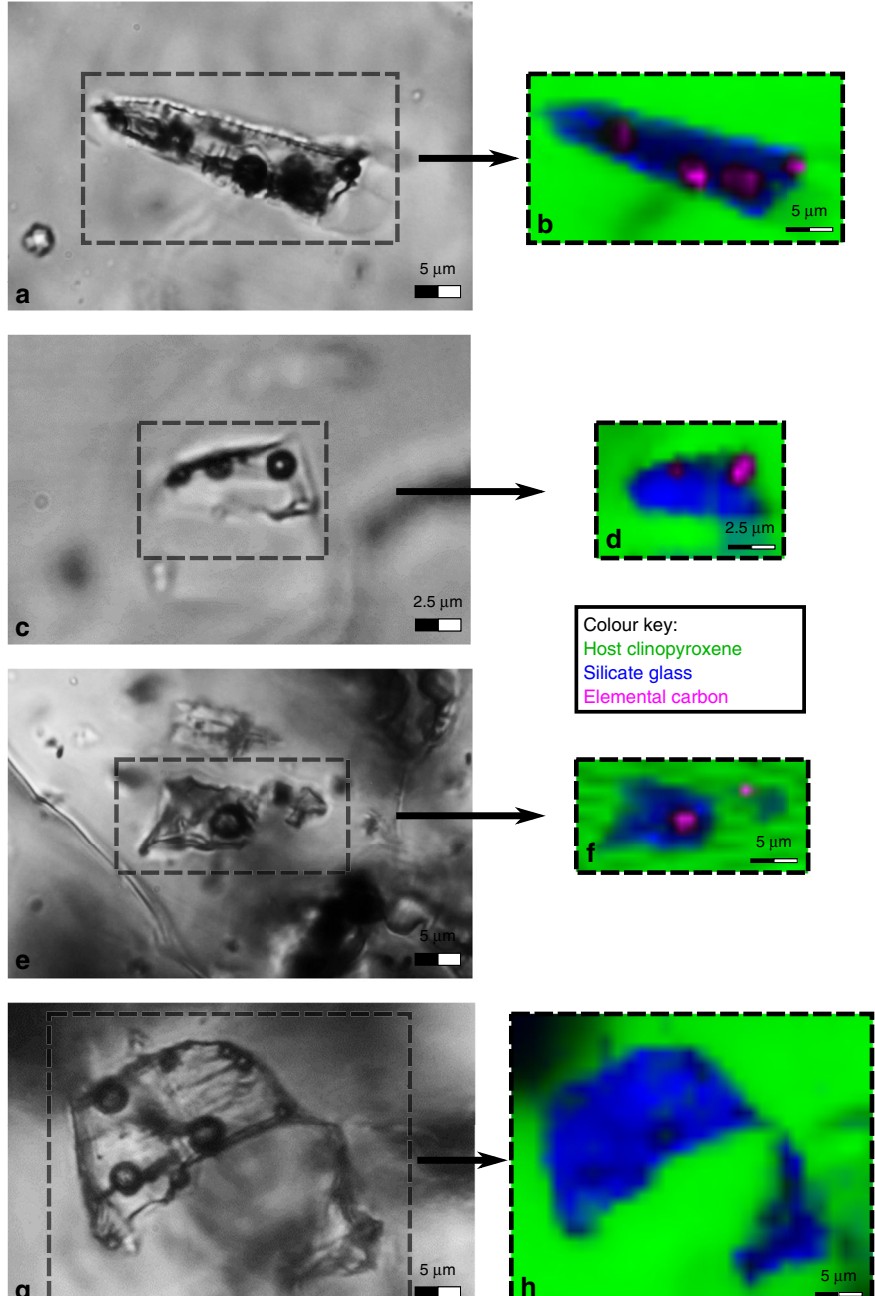

**Fig. 4 Bubble-bearing melt inclusions at transmitted light optical microscopy and confocal Raman microspectroscopy.** Left column: transmitted light photomicrographs at optical microscope of the analysed areas, bordered by dotted lines. Right column: Raman hyperspectral maps of the corresponding areas. **a**, **c**, **e** Photomicrographs of elemental carbon-bearing single- and multi-bubble MIs (**a** and **c**: sample NEW31, New Jersey, USA; **e**: sample AN39, Morocco). **b**, **d**, **f** Raman hyperspectral maps of the same samples area. **g** Photomicrograph of an irregular-shaped $CO_2$-bearing multi-bubble MI (sample NS12, Nova Scotia, Canada). **h** Raman hyperspectral map of the same sample area. The Raman signal of $CO_2$ is weak due to its low density. However, spot analyses confirmed the presence of $CO_2$ in all bubbles.

host clinopyroxene crystal clots can be calculated from mineral compositions (Supplementary Table 6) using methods developed for magmatic systems[37,38] (Supplementary Note 2). The geothermobarometer based on the equilibrium between clinopyroxene and a magmatic liquid[37] was applied using whole rock composition as proxy for the original magmatic liquid composition, because the MIs glass is in chemical disequilibrium with the host clinopyroxene. The calculated crystallization pressure ranges from 0.1 to $0.7 \pm 0.2$ GPa (at temperatures from 1150 to $1230 \pm 27$ °C) and is consistent with previous estimates from clinopyroxene crystallization pressures (from 0.2 to 0.8 GPa) in basalts

from the entire CAMP[23,39–41]. These results suggest that the crystallization of clinopyroxene in the investigated CAMP samples occurred predominantly within the middle continental crust (on average ca. $12 \pm 7$ km for a pressure/depth gradient of about 0.03 GPa/km; Supplementary Fig. 7).

The deep origin of MIs is consistent with observed volatile concentrations in both their glass and bubbles (Supplementary Note 3). The presence of sulphides within some MIs shows that the entrapped melt became sulphide-saturated with S concentrations likely exceeding 1500 ppm[42,43]. Sulphur concentrations of the same order of magnitude were estimated for CAMP

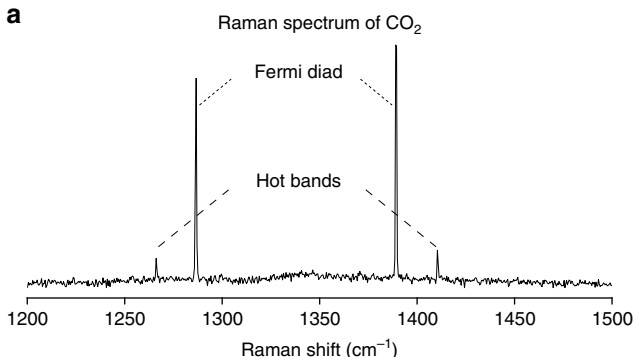

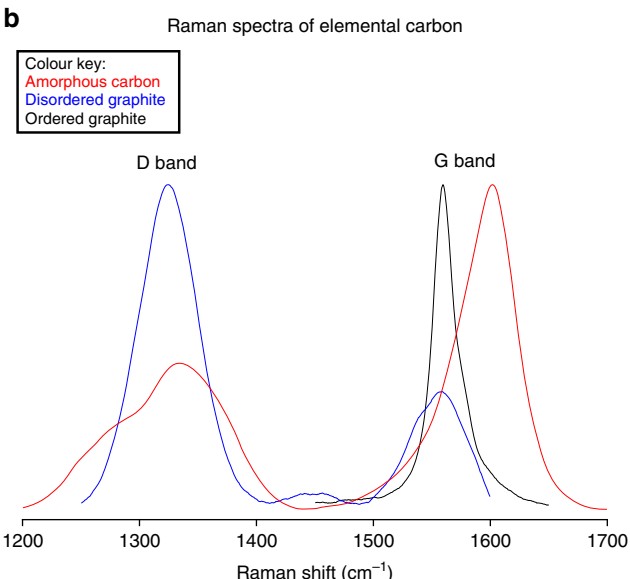

**Fig. 5 Raman spectra of CO$_2$ and elemental carbon. a** Raman spectrum of CO$_2$, acquired on sample NS9 (Nova Scotia, Canada). The Fermi diad is represented by sharp bands, at 1285 cm$^{-1}$ and at 1388 cm$^{-1}$, and the hot bands are represented by symmetrical weak bands, below 1285 cm$^{-1}$ and above 1388 cm$^{-1}$. **b** Raman spectra of elemental carbon: amorphous carbon acquired on sample NEW31 (New Jersey, USA), disordered graphite acquired on sample AN39 (Morocco), and ordered graphite (detail on the G band) acquired on a common pencil. Compared to the ordered graphite Raman spectrum, our Raman spectra of disordered graphite and amorphous carbon always have one or more D peaks between 1200 and 1400 cm$^{-1}$. The G band lower than 1590 cm$^{-1}$ indicates disordered graphite, while the G band higher than 1590 cm$^{-1}$ indicates amorphous carbon (see "Methods" section). The D band is often composed by two peaks (D1 at ca. 1350 cm$^{-1}$ and D5 at ca. 1270 cm$^{-1}$) for both disordered graphite and amorphous carbon.

basalts[44]. Moreover, about 0.5–0.6 wt% H$_2$O was detected in the MIs glass through NanoSIMS analysis, revealing hydrated conditions for these melts. Despite the presence of H$_2$O and S in the MIs glass, these volatiles were not detected in the bubbles. Hence, considering a realistic maximum primary concentration of ca. 1 wt% H$_2$O and ca. 0.1 wt% SO$_2$ in tholeiitic within-plate basaltic melts[44,45], most H$_2$O and SO$_2$ are expected to exsolve at pressures lower than 0.1 GPa (i.e., <3 km depth)[16,46]. Even considering that H$^+$ may move from the bubbles into the glass, and CO$_2$ from the glass into the bubbles after MI entrapment[47,48], the observed distribution of volatile species between glass and bubbles within MIs suggests the dominant occurrence of gas exsolution and bubble formation at relatively high pressures from a CO$_2$-rich melt.

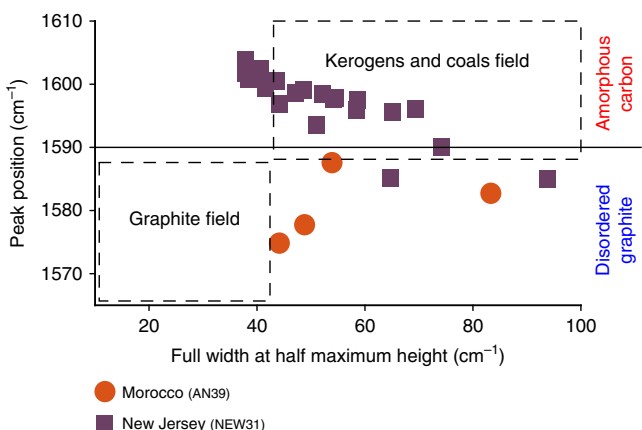

**Fig. 6 Crossplot of the Raman spectra of elemental carbon.** This crossplot displays the Raman spectra of elemental carbon with the peak position of the G band ranging from ca. 1575 cm$^{-1}$ to ca. 1605 cm$^{-1}$. The line in correspondence of 1590 cm$^{-1}$ peak position value separates the disordered graphite data (below) from the amorphous carbon data (above) according to the present study (see "Methods" section). The areas bordered by dashed lines distinguish the graphite field from the kerogens and coals field according to ref. [34]. The error on PP and therefore also on FWHM is considerably smaller than the spectral resolution for the Raman spectra displayed in the crossplot (0.8 cm$^{-1}$)[70], and thus is much smaller than the plotted symbols.

The inferred depth of CO$_2$ exsolution and entrapment indicates that this volatile species has a deep origin (at least 12 ± 7 km on average). It therefore reveals that the entire CO$_2$ budget involved in CAMP emplacement could not have originated exclusively from assimilation and degassing of shallow intruded sediments[26], because sediments in the circum-Atlantic basins only reach a thickness of 5 km in eastern North America[49] and <1 km in Morocco[23] and Portugal[39]. On the contrary, at least part of the CO$_2$ most probably derived from assimilation of deep- to middle-crustal metasedimentary rocks (e.g., metacarbonates or graphite-bearing amphibolites/granulites) or from the mantle source of CAMP basalts (Fig. 7), containing significant amounts of recycled sedimentary material[23,39,40,50,51].

The calculated depth of entrapment (ca. 12 ± 7 km) allows an estimation of the CO$_2$ concentration originally present in CAMP magmas. The CO$_2$ saturation in basaltic melts is achieved at ca. 1000 ppm at 0.2 GPa, increasing by ca. 500 ppm for each 0.1 GPa[52]. Considering the calculated crystallization depths, the minimum estimate for the CO$_2$ concentration of CAMP magma, before gas exsolution, is between ca. 500 and 4000 ppm. Such values are consistent with the CO$_2$ concentrations in the MIs, calculated from CO$_2$ density within the bubbles, which range from 0.5 to 1.0 wt%. Moreover, starting from the minimum calculated values of the CO$_2$ concentration within MIs (i.e., 0.5–0.6 wt%) as representative of CAMP magma, assuming an average density of 2.90 g/cm$^3$ for basaltic rocks[53] and considering 5–6 × 10$^6$ km$^3$ for the total volume of CAMP (in order to take into account the deep plumbing system), the total amount of degassed volcanic CO$_2$ during CAMP emplacement would be up to 10$^5$ Gt. Interestingly, the values estimated for the CO$_2$ concentration of CAMP magma (0.5–1.0 wt%) and for the total amount of degassed volcanic CO$_2$ during CAMP emplacement (up to 10$^5$ Gt) are consistent with those assessed in several other LIPs, using different approaches[29].

**Implications for the end-Triassic climatic and environmental changes.** The high-volume fractions of CO$_2$- and elemental

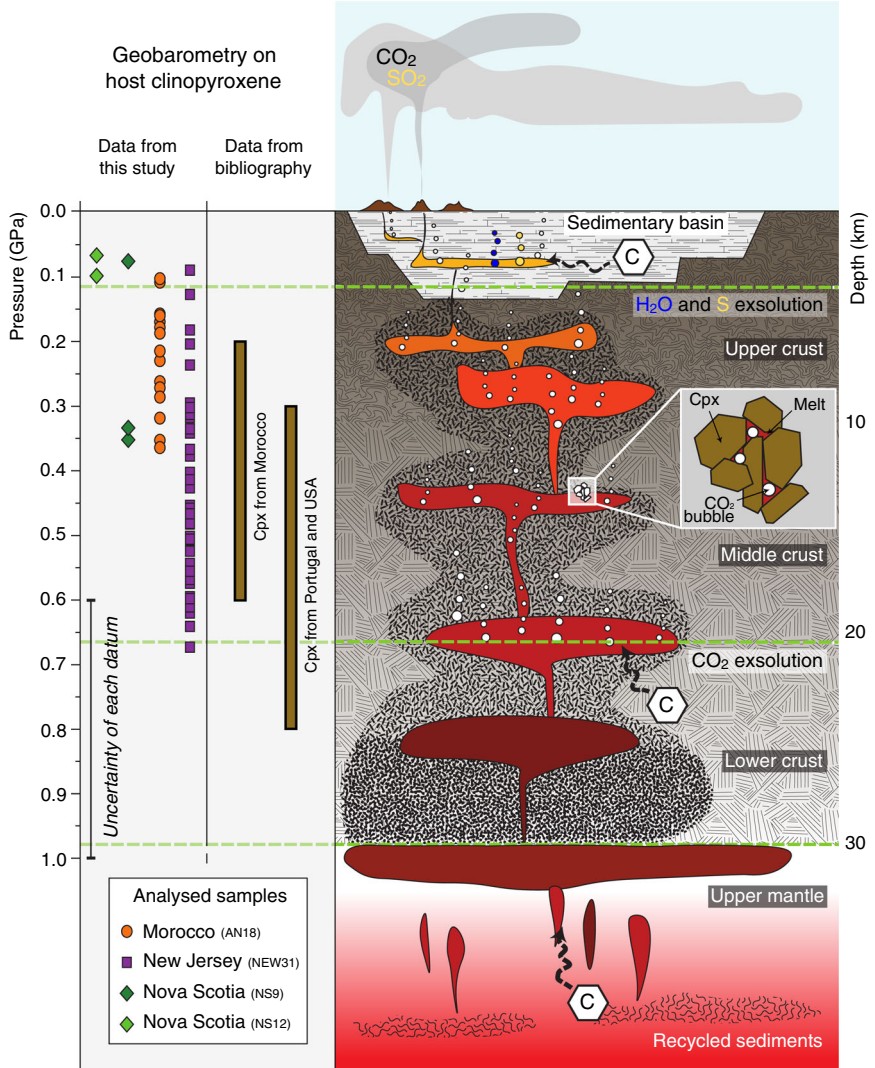

**Fig. 7 Sketch of the transcrustal plumbing system of CAMP basaltic magmas from the mantle to the surface.** The evolution of basaltic magmas occurs at variable depth by crystallization of minerals, which then form aggregates in crystalline mushes[13,14] and entrain bubble-bearing melt, forming MIs. Different volatile species exsolve at variable depth[16]. In particular, $CO_2$-rich fluids (white bubbles) start exsolving at great depth, whilst $H_2O$-rich fluids (blue bubbles) and S-rich fluids (yellow bubbles) start exsolving at shallow depth. The black dashed arrows indicate the potential sources for the carbon in CAMP magma: the mantle, the deep crust and the Palaeozoic or Triassic sedimentary basins in which CAMP sills intruded. The carbon within the here studied MIs derives from the deep sources as demonstrated with clinopyroxene geobarometry data. Clinopyroxene crystallization pressures of this study have been calculated using ref. [37] (Supplementary Note 2). Clinopyroxene crystallization pressures of bibliography are from ref. [23] for Morocco, ref. [39] for Portugal, and ref. [40] for USA. The error (±0.2 GPa) takes into account the uncertainties from both the geobarometry model (±0.1 GPa)[37] and the electron microprobe analyses (±0.1 GPa, deriving from the ±10% accuracy on measured Na concentration).

carbon-bearing bubbles within CAMP MIs, along with the inferred depths of formation, reveal the high abundance (0.5–1.0 wt%) of $CO_2$ in the CAMP transcrustal magmatic plumbing system. The $CO_2$-bearing bubbles identified in CAMP MIs can be interpreted as batches of ascending volatiles entrapped in crystalline mush shortly prior to its mobilization and prior to eruption. This evidence for $CO_2$ saturation in the basaltic magmas at depth can explain the pulsed eruptive style of CAMP, where $CO_2$ acts as propellant for magma ascent, causing rapid and violent eruptive pulses. For instance, $CO_2$-rich Hawaiian basalts have been shown to rapidly rise from over 5 km depth and to cause high fountaining eruptions[54].

The presence of large amounts of $CO_2$-bearing bubbles, the pulsed eruption and the efficient degassing of $CO_2$ from the basaltic magmas[55], strengthens the role of CAMP in triggering end-Triassic extreme greenhouse conditions[56]. The rate of volatile release plays a fundamental role in determining the severity of the surface environmental response; more rapid release increases the maximum transient concentration of atmospheric $CO_2$ and the subsequent severity of any environmental cascade. Assuming 0.5–1.0 wt% $CO_2$ in CAMP basalts, as suggested by the average $CO_2$ density and most common glass/bubble ratio within MIs, and considering its efficient rise to the atmosphere through the magmatic transcrustal roots, it is possible that just a single CAMP volcanic pulse may have severely affected the end-Triassic climate. In fact, a single short-lived CAMP magmatic pulse (ca. $10^5$ km$^3$ erupted over 0.5 kyrs)[21–23] may emit about $5 \times 10^{16}$ mol $CO_2$, roughly the same total amount of projected anthropogenic emissions over the 21st century, according to the Representative Concentration Pathway 4.5[57]. This scenario for rapid $CO_2$ emissions predicts a global temperature increase of about 2 °C and an oceanic pH decrease of about 0.15 units over 0.1 kyrs, and

suggests that the end-Triassic climatic and environmental changes, driven by $CO_2$ emissions, may have been similar to those predicted for the near future.

## Methods

**Sample selection and preparation**. The basaltic rocks analysed for the present study were sampled from CAMP lava flows and sills in North America (USA and Canada), Africa (Morocco), and Europe (Portugal; Supplementary Table 1). Using reflected and transmitted light optical microscopy, samples with bubble-bearing MIs were selected for this study from a total of over 200 intrusive and effusive samples screened from throughout CAMP. Mainly basalt lava flows were selected for this study, as they appear richer in MIs than intrusive rocks. Because of the possible contaminations and the very low carbon concentrations, characterizing the volatile content of bubbles in MIs within 201 Ma-old basaltic rocks is extremely challenging. In order to prevent any potential contamination of carbon species while preparing, analysing, and handling the samples, steel tools and organic compounds, usually involved in sample cut and polishing (e.g., cyanoacrylate glue and Canada balsam), were carefully avoided or completely dissolved in acetone before analysis. Both thin and thick sections were prepared. In particular, glue-free double-polished thick (about 100 µm) sections were used for in situ confocal Raman microspectroscopy, in order to avoid contamination and signal interferences from any carbon-bearing organic compounds.

**Confocal Raman microspectroscopy analysis**. Confocal Raman microspectroscopy was employed to detect and analyse the solid and fluid phases within bubbles of unexposed MIs, and to analyse the glass of exposed MIs. Both thin and thick sections were used. In the glue-free double-polished thick sections, analyses were carried out below the sample surface for all the phases within bubbles of unexposed MIs, and on the sample surface for the glass of exposed MIs. The analyses were conducted at the Research and Instrument Core Facility of the Faculty of Science, Eötvös Loránd University of Budapest, using a HORIBA JobinYvon LabRAM HR 800 Raman microspectrometer. This analytical technique was applied to all samples involved in this study, for the characterization of carbon species within bubble-bearing MIs. Both spot and areal analyses were carried out. Spot Raman analysis allowed us to acquire spectra of the phases present in glass and bubbles of MIs, and to investigate their crystalline form (for solid phases) and their density and pressure (for fluid phases) through spectral features. Areal Raman analysis (i.e., Raman mapping) allowed us to reconstruct the spatial distribution of solid and fluid phases in bubbles of MIs. A frequency doubled Nd-YAG green laser with a 532 nm excitation wavelength was employed, displaying 120 mW at the source and 23 mW on the sample surface, and an OLYMPUS ×100 objective was used to focus the laser on the analysed sites. Raman spectra acquisition, in both single- and multi-window settings, was conducted at room temperature, using a 100 µm confocal hole (50 µm confocal hole for maps on NEW31), 1800 grooves/mm optical gratings (600 grooves/mm optical gratings for all the maps and some spot spectra), 2–10 accumulations and 8–120 s exposition time. Furthermore, a He–Ne red laser with a 633 nm excitation wavelength was occasionally used in order to distinguish fluorescence emissions. The investigated spectra range from 100 to 4000 $cm^{-1}$, depending on the spectral region of interest for each analysed phase. The spectral resolution of measurements varied from 0.8 to 3.0 $cm^{-1}$ for the spot spectra, and from 2.4 to 3.0 $cm^{-1}$ for the maps. The high degree of variability in analytical conditions is due to the very different types of analysed materials, and to the general low quantity of the analysed phases within bubbles of MIs. All the Raman data were processed through LabSpec 5 and OMNIC For Dispersive Raman softwares.

**Electron microprobe (EMP) analysis**. EMP was employed to analyse the chemical composition (major, minor and some trace elements) of the glass in exposed MIs and their host clinopyroxene crystals. The analyses were conducted at the C.N.R., Institute of Geosciences and Georesources in Padova, using a Cameca SX50 EMP (samples AL14, AN18, NEW31, and NS12), and at the Department of Earth Sciences, University of Milano, using a JEOL JXA 8200 Superprobe (samples AN137A, NS9, and NS12). As indicated by replicate analyses of standards and unknown samples at both laboratories, the analyses conducted at Padova and Milano are equivalent and comparable, for the level of accuracy required in the present study (Supplementary Tables 2 and 6). Natural and synthetic standards were used for instrumental calibration. For the measurement of the glass in exposed MIs, a 1 µm beam diameter was used to avoid contamination from the surrounding host minerals. Standard beam current and accelerating voltage conditions were applied (Supplementary Tables 2 and 6).

**Scanning electron microscopy with energy-dispersive X-ray spectroscopy (SEM–EDS) analysis**. SEM–EDS was employed to semi-quantitatively estimate the major element composition of the glass in exposed MIs and to chemically map the glomerocrystic clinopyroxene aggregates surrounding the MIs, in terms of Na, Mg, Al, Si, K, Ca, Ti, Cr and Fe. The analyses were conducted at the Department of Geosciences, University of Padova, using a CamScan MX3000 SEM, with a $LaB_6$ source, equipped with an EDAX spectroscopy probe (sample NEW31). Both spot

and areal analyses were carried out in order to qualitatively distinguish chemically different phases and to detect possible compositional variations.

**Nanoscale secondary ion mass spectrometry (NanoSIMS) analysis**. NanoSIMS was employed to quantify the volatile concentration of the glass in exposed MIs (in terms of $CO_2$ and $H_2O$). The analyses were conducted at the Muséum National d'Histoire Naturelle in Paris, using a Cameca NanoSIMS 50 (samples NEW31 and NS21). Thin section chips with exposed glass of MIs were attached to a clean Al disk with double-sided Cu tape, and Au coated (20 nm thick). In order to remove surface contamination, in addition to the coating, and to reach a steady-state sputtering regime before each analysis[58], the sample surface was presputtered for 150 s by a $Cs^+$ primary beam set at 170 pA and rastered over $5 \times 5$ $\mu m^2$ area. For data acquisition, the $Cs^+$ primary beam current was set at 15 pA. While the beam was rastered over $3 \times 3$ $\mu m^2$ area, only ions from the inner $1 \times 1$ $\mu m^2$ were collected using the beam blanking mode, in order to reduce contamination from the surface and the surrounding phases. Each analysis comprises a stack of 100 cycles, each cycle being 1.024 s long. Secondary ions of $^{12}C^-$, $^{16}OH^-$, and $^{28}Si^-$ were acquired simultaneously in multicollection mode by electron multipliers with a dead time of 44 ns. Mass resolving power was set at 6000 in order to resolve potential interferences on $^{16}OH^-$. During the measurement session, the vacuum in the analysis chamber remained below $3 \times 10^{-10}$ Torr. The $CO_2$ and $H_2O$ concentrations of MIs glass were obtained through the measurement of $^{12}C^-/^{28}Si^-$ and $^{16}OH^-/^{28}Si^-$ ratios, respectively. These ratios were subsequently converted into concentrations through calibration curves, determined using standards of known compositions, in particular B and STR standards for both $CO_2$ and $H_2O$[59,60]. Concentrations and uncertainties were calculated using the R programme, considering both counting statistics on each analysis and uncertainties of the calibration curves.

**Carbon dioxide analysis**. Because of resonance effect, the Raman spectrum of $CO_2$ is characterized by two sharp bands, at ca. 1285 $cm^{-1}$ and at ca. 1388 $cm^{-1}$, usually called Fermi diad or Fermi doublet, associated to two symmetrical weak bands, below 1285 $cm^{-1}$ and above 1388 $cm^{-1}$, usually called hot bands[33] (Fig. 5a and Source Data 1). During the last few decades several $CO_2$ densimeters have been proposed[61–67]. These tools are based on the systematic dependence between the Fermi diad splitting in $CO_2$ spectra (i.e., the formation of two split bands, due to the Fermi resonance phenomenon) and the density of $CO_2$[33]. In the present study, all analysed $CO_2$-bearing bubbles display a constant value of the Fermi diad splitting, corresponding to 102.6–102.8 $cm^{-1}$ (Supplementary Table 4). Applying the $CO_2$ densimeter of ref. [62], which is optimal for basaltic magmas[27], these values correspond to low densities of about 0.1 $g/cm^3$ (Supplementary Table 4). Moreover, assuming 0.1 $g/cm^3$ for $CO_2$ density within all analysed bubbles (calculated with the densimeter of ref. [62]), 2.75 $g/cm^3$ for MI glass density[27] and 0.1–0.2 (up to >0.5 exceptionally) volume fraction of $CO_2$-rich bubbles within each MI (optically estimated for samples AN18, NS9 and NS12), the $CO_2$ concentration within the whole MI (i.e., glass plus bubbles) ranges from 0.5 to 1.0 wt%.

**Elemental carbon analysis**. The Raman spectrum of elemental carbon is characterized by a first and a second order[34,68]. In correspondence of the first-order spectrum, two different bands are present: the D band (disordered/defect-activated) at ca. 1350 $cm^{-1}$ and the G or O band (graphite or ordered) at ca. 1580 $cm^{-1}$ (Fig. 5b and Source Data 2). The D band is a composite one, composed of five different peaks (from D1 to D5, ranging from ca. 1200 $cm^{-1}$ to ca. 1700 $cm^{-1}$ approximately). These peaks, in the first-order spectrum, are activated in disordered graphite by lattice defects and are typical of non-crystalline structures[34,35]. Because of the many different features depending on lattice defects of low-crystalline materials, the D band is very difficult to interpret. The G band is a single sharp peak, typical of graphite, and is a good indicator for the degree of crystallinity of any analysed material[34]. For this reason, the G band is usually employed in crossplots, the graphical representation of peak features, where the peak position of the Raman shift (PP) is plotted versus the full width at half maximum height (FWHM). In correspondence of the second-order spectrum, one band is present: the 2D band or G′ at ca. 2700 $cm^{-1}$. The 2D band is absent or very weak for non-crystalline materials[34]. In this study, we used a crossplot for the spectra of elemental carbon displaying the peak position of the G band ranging from ca. 1575 $cm^{-1}$ to ca. 1605 $cm^{-1}$, in order to distinguish the different types of elemental carbon (Fig. 6). For non-crystalline materials the G band is shifted to higher values of wavenumber (i.e., Raman shift) and is wider than the normal band and is at ca. 1580 $cm^{-1}$ (ref. [34]). However, the ubiquitous presence of the D band in all our Raman spectra indicates that, also for PP values typical of graphite (i.e., ca. 1580 $cm^{-1}$), elemental carbon shows lattice defects and general disorder, and therefore it is present as disordered graphite. Here, PP values lower than 1590 $cm^{-1}$ are considered indicative of disordered graphite, while PP values higher than 1590 $cm^{-1}$ are considered indicative of amorphous carbon (Fig. 6). In detail, sample AN39 contains only disordered graphite, instead sample NEW31 contains both disordered graphite and amorphous carbon. Furthermore, our data present a wide range for both PP values, indicating materials with different crystallinity, and FWHM values, indicating materials with different thermal maturity. In particular, the FWHM values usually above 40 $cm^{-1}$ reveal a general low-thermal maturity of the elemental carbon (as expected for materials of poor crystallinity).

**MIs glass analysis**. The glass in exposed MIs was investigated by EMP (samples AL14, AN18, AN137A, NEW31, NS9 and NS12), by SEM–EDS (sample NEW31) and by NanoSIMS (samples NEW31 and NS21). Preliminary confocal Raman microspectroscopy analysis (samples AN39, AN137A, NEW31 and NEW73) revealed the presence of $H_2O$ within the glass. The Raman spectrum of glass shows the wide band of $H_2O$, at ca. 3600 $cm^{-1}$, and some broad bands, mainly at ca. 500 $cm^{-1}$ and ca. 750 $cm^{-1}$, which is distinguishable from the strong Raman spectrum of the surrounding host mineral and are common for silicate glasses[69] (Supplementary Fig. 6). The presence of volatile species within the MIs glass, also supported by the relatively low totals of EMP analysis (generally around 97–99 wt%; Supplementary Table 2), is confirmed by NanoSIMS. This analysis shows that $H_2O$ concentration ranges between 0.5 and 0.6 wt%, and $CO_2$ concentration ranges between 30 and 90 ppm (Supplementary Table 5). The majority of MIs has andesitic composition, clearly different from typical CAMP basalts or basaltic andesites (Supplementary Fig. 3). MIs glass is generally enriched in $SiO_2$ and $Al_2O_3$, and depleted in FeO (total Fe), MgO, and CaO compared to the host rocks, and its composition shows roughly correlated major element variations (Supplementary Fig. 4). Two exceptions are the glasses analysed in two MIs from samples AL14 and AN18 showing trachyte and basaltic andesite compositions, respectively (Supplementary Fig. 3). In order to estimate the amount of fractionation necessary to evolve from an average CAMP basalt composition to an average MIs glass composition mass balance calculations were developed. We considered the measured glass of an exposed MI in sample NEW31 (from this study) as the MIs glass composition, the whole-rock composition of sample NEW31 (from ref. [40]) as a typical CAMP basalt, and measured mineral compositions (from this study and ref. [23]). These calculations suggest that the average composition of the MIs glass can be obtained by ca. 55% crystallization of a typical CAMP basalt, where the fractionating assemblage consists of ca. 39% augitic clinopyroxene, 11% plagioclase and 5% magnetite. The discrepancy between the small amount of observed crystallization (of clinopyroxene in thin films, and of magnetite in tiny crystals) versus estimated crystallization and the lack of plagioclase within the MIs suggest that at least part of the crystallization, necessary to explain the differentiated nature of the MIs glass, occurred before entrapment. Hence, the differentiation process of MIs glass is not exclusively due to post-entrapment crystallization and diffusive re-equilibration.

**Host clinopyroxene analysis**. The host clinopyroxene was investigated by EMP (samples AL14, AN18, AN137A, NEW31, NS9 and NS12) and by SEM–EDS (sample NEW31) for major and minor element compositions (transects and mapping). All the compositional transects on the host clinopyroxene were conducted by EMP from the rim of an exposed MI to the core of the host mineral aggregate, using a step size of ca. 3–4 μm between analysis points with ca. 1 μm beam diameter (Supplementary Fig. 5 and Supplementary Table 6). The host glomerocrystic clinopyroxene aggregates have an augitic composition, but some aggregates show a decrease in CaO content and a slight increase in both MgO and FeO contents close to the MIs (Supplementary Fig. 5). The presence of a thin rim around the MIs of slightly Ca-depleted and Fe ± Mg-enriched clinopyroxene compared to the augitic composition at distance from the MIs in glomerocrystic aggregates, suggests the probable presence of augite–pigeonite exsolution lamellae close to the contact with the entrapped melt. Moreover, the SEM–EDS mapping of host glomerocrystic clinopyroxene aggregates revealed a compositional zoning and the presence of both augite and pigeonite in these aggregates (Fig. 3). Faint halos and local regions of relatively low Ca and high Fe ± Mg around the MIs in SEM–EDS maps support the presence of augite–pigeonite exsolution lamellae close to the MIs glass. Therefore, it is likely that such augite–pigeonite mixtures around MIs formed at subsolidus conditions from an intermediate composition clinopyroxene, which crystallized from the melt after its entrapment (i.e., post-entrapment crystallization). Since such features could not be detected by EMP analyses, it is suggested that the crystallized MIs rim is less than ca. 3–4 μm in thickness and the exsolution lamellae are narrower than ca. 1 μm.

## Data availability

All data generated and discussed in this study are presented in full in the Supplementary Information (Supplementary Notes 1–3, Supplementary Figs. 1–7, Supplementary Tables 1–6 and Supplementary References), and in the Source Data 1 (for the Raman spectra of $CO_2$) and Source Data 2 (for the Raman spectra of elemental carbon).

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

## Acknowledgements

This study was supported by the following collaborative research projects: PRIN 20178LPCP (Italy) to A.M. and R.S., CNRi (Italy)-CNRST (Morocco) to A.M. and N.Y., Mega-Grant 14. Y26.31.0012 (Russian Federation) to N.Y., ELTE Institutional Excellence Programme 1783-3/2018/FEKUTSRAT (Hungary) to L.E.A. and Cs.Sz., NERC Large Grant NE/N018559/1 (UK) to R.J.N., Grant SIR RBSI14Y7PF (Italy) to O.B., National Science and Engineering Research Council of Canada to D.R.B., CNRS, Région Île-de-France, Ministère délégué à l'Enseignement supérieur et à la Recherche, and Muséum National d'Histoire Naturelle to L.R. (NanoSIMS facility, Muséum National d'Histoire Naturelle, Paris, France). The authors thank L. Tauro (University of Padova) for the sample preparation, R. Carampin (C.N.R., I.G.G., Padova) and A. Risplendente (University of Milano) for EMP analyses, O. Gianola (University of Padova) for NanoSIMS analyses, and K. Putirka (California State University) for providing useful information on clinopyroxenes geothermobarometry. H. Bertrand, L. Tanner, D. Kontak and many others are thanked for assistance during field-work.

## Author contributions

A.M., S.C. and J.D.C. devised the project. A.M., S.C. and N.Y. carried out the field-work. M.C., A.M., L.E.A., L.R. and R.S. collected the analytical data. M.C., A.M., L.E.A., S.C., J. D.C., R.J.N., B.J.W.M., P.B.W., O.B., D.R.B., N.Y., L.R., R.S. and Cs.Sz. contributed to the multidisciplinary aspects and writing of the manuscript.

## Competing interests

The authors declare no competing interests.
