## [Peer Review File · Nature Communications]

Reviewers' comments (received on 27th November 2019)

Reviewer #1 (Remarks to the Author):

This is an exciting and rigorous investigation of the CO₂ contents of magmas erupted during one of Earth's largest volcanic events; the end Triassic Central Atlantic Magmatic province. Determining the volatile contents of such magmas is challenging and the authors have undertaken a novel approach to calculating CO₂ contents. By selecting specific samples from a large number of samples (200) of basalt collected across this vast igneous province Capriolo et al. have been able to find relatively pristine melt inclusions with bubbles of CO₂. Using confocal Raman microscopy, the authors show that the inclusions contain bubbles of both elemental C and CO₂. Distinguishing between the different types of carbon and identifying the sources as graphite or kerogens/coals is an exciting and novel part of this study (although may be not emphasized enough) because previous work (e.g. Ganino & Arndt, 2009) has suggested that sedimentary sourced carbon is a major source of CO₂ outgassed by flood basalts erupted in Siberia and responsible for the End Permian mass extinction event.

Specific Comments

1. Title. It would be good to include 'Central Atlantic Magmatic Province'. May be change this to 'Deep CO₂ in the end-Triassic Large Central Atlantic Magmatic province'?
2. Line 42. There needs to be some mention that the study focuses on flood basalts, which volumetrically dominate Large Igneous Provinces, rather than the spatially & temporally associated alkaline igneous rocks (carbonatites etc).
3. Line 51. A mention to the work of Karlstrom & Richards (2011) would seem appropriate here (and elsewhere in the manuscript). This was the first paper to quantitatively discuss the role of CO₂ during high-pressure fractionation of clinopyroxene in lower crustal magma chambers believed to be associated with many flood basalt plumbing systems.
4. Line 69. Issues related to gas exsolution bubbles within melt inclusions in flood basalts were discussed by Black & Gibson ((2019), pg 320). I appreciate this paper may have been published very shortly before the authors submitted their manuscript.
5. Line 78. Change 'slightly' to 'minor'.

6. Line 94. Change 'suggest' to 'suggests'

7. Line 153. Please supply the analyses of the host clinopyroxenes and plagioclase feldspars. I'm assuming that the clinopyroxenes are sub-calcic augites as proposed to crystallise at high pressures in the experimental study of flood basalts by Thompson (1974)? Regardless, the reader is not able to calculate the crystallisation depths of the clinopyroxenes (using the method of Putirka etc) without this information. To do this, the composition of the clinopyroxene that hosts each melt inclusion needs to be clearly defined in a table. This information is required to calculate pressure and vital to calculating the CO₂ contents of the melt inclusions (Line 153).

8. Line 159. Please add information to the main text related to how the total estimate of 10⁵ Gt for the Central Atlantic Magmatic Province was made. This needs to include some of the following details that were provided post submission:

'Starting from the minimum calculated values of the CO₂ concentration in our melt inclusions (i.e., 0.5-0.6 wt. %) as representative of CAMP magma, we evaluated the total amount of degassed volcanic CO₂ during CAMP emplacement (i.e., up to 10⁵ Gt). For this calculation, we considered an average density of 2.90 g/cm³ for basaltic rocks and 5-6 x 10⁶ km³ for the total volume of CAMP, in order to take into account also the deep plumbing system ("Over 2 x 10⁶ km³ of CAMP magmas" is reported at line 60 of the Main Text)'

9. The total estimate of 10⁵ Gt erupted over 400,000 yrs is similar to the 3000 Mt per year estimated by Black & Gibson (2019) and merits some discussion.

10. Lines 156 to 160. The concentrations of CO₂ (0.5 to 1 wt. %) that the authors calculate for the melts trapped in clinopyroxene from the CAMP are almost identical to (i) those estimated for the North Atlantic Province and Siberian Traps and (ii) CO₂ estimates predicted for the amounts of melting associated with the generation of tholeiitic magmas in upwelling mantle plumes (see Black & Gibson, 2019). These studies involved different approaches (volatiles versus trace element ratios of melt inclusions, numerical melt models) but it is surely noteworthy that the findings from the study by Capriolo et al are consistent with those estimated for flood basalts erupted in different Large Igneous Provinces.

In summary, this is a well-written and nicely illustrated manuscript. The work is innovative, the data high quality and the interpretations are exciting. The paper will be of great interest to the readers of Nature Communications and, subject to the minor points above being addressed, I believe that it is worthy of publication.

Sally A Gibson

21st November 2019

References

Black, B.A., Gibson, S.A., 2019. Deep Carbon and the Life Cycle of Large Igneous Provinces. *Elements* 15, 319–324.

Karlstrom, L., Richards, M., 2011. On the evolution of large ultramafic magma chambers and timescales for flood basalt eruptions. *J. Geophys. Res.* 116, 13 PP. <https://doi.org/201110.1029/2010JB008159>

Thompson, R.N., 1974. Primary basalts and magma genesis. *Contrib. Mineral. Petrol.* 45, 317–341.

Reviewer #2 (Remarks to the Author):

Deep CO₂ in the end-Triassic Large Igneous Province

I am happy to provide an opinion as originally requested on the significance of the advance of the authors study and the links to the end-Triassic extinction event.

This paper uses new observations on the contents of gas exsolution bubbles contained in melt inclusions within mainly pyroxene crystals, in order to argue for large quantities of carbon-dioxide from deep crustal and subcrustal sources in large igneous province melts. The evidence is very clearly set out and coherently argued and, if correct, represents an important contribution to our understanding of the relationship between large igneous province formation and mass extinctions and/or extreme climate change. This paper is particularly significant because it focusses one of the largest but least well understood mass extinctions in the Phanerozoic, in the latest Triassic. The real advance here would be viability of deep carbon reservoirs (and presumably other deeply sourced volatiles) as the principal driver of environmental change, rather than the shallow sources which are currently the main focus of research.

This is a multidisciplinary study and I do not feel well qualified to provide comment on the deep crustal or mantle processes discussed, nor on the details of the analytical methods – it is important that these aspects of the paper are separately reviewed. However, the basic observations do seem to me to be strong, and also based on analysis of material from a sufficiently wide range of samples from different parts of the the Central Atlantic Magmatic Province in order to allow the broader conclusions to be reached.

My main criticism of the paper as written is the balance between material in the main text and the supplement. The results section (lines 75–104) makes repeated reference to Supplementary Information (tables and figures), which I personally found frustrating. These results are the bedrock on which the interpretations are based, and should be a central part of the main text. On the other hand, figures in the main text (Figure 3) include examples of the Raman spectra which, although also important, is the kind of more detailed evidence that may be best placed in supplementary data. Additionally, Figs 1 and 2 contain multiple examples of essentially the same phenomena. I recommend, then, that the balance of material in the main text and the supplementary information is reconsidered, such that the main observational results are fully illustrated in the main text.

Capriolo et al., “Deep CO₂ in the end-Triassic Central Atlantic Magmatic Province”

Response to reviewers' comments.

All comments are reproduced below and our responses follow in bold text. The numbers of lines in the text are referred to the revised version, where corrections are tracked.

Reviewer #1 (Remarks to the Author):

This is an exciting and rigorous investigation of the CO₂ contents of magmas erupted during one of Earth's largest volcanic events; the end Triassic Central Atlantic Magmatic province. Determining the volatile contents of such magmas is challenging and the authors have undertaken a novel approach to calculating CO₂ contents. By selecting specific samples from a large number of samples (200) of basalt collected across this vast igneous province Capriolo et al. have been able to find relatively pristine melt inclusions with bubbles of CO₂. Using confocal Raman microscopy, the authors show that the inclusions contain bubbles of both elemental C and CO₂. Distinguishing between the different types of carbon and identifying the sources as graphite or kerogens/coals is an exciting and novel part of this study (although may be not emphasized enough) because previous work (e.g. Ganino & Arndt, 2009) has suggested that sedimentary sourced carbon is a major source of CO₂ outgassed by flood basalts erupted in Siberia and responsible for the End Permian mass extinction event.

We are grateful for the reviewer's positive assessment and have made all of the changes that have been requested. We thank the reviewer for her time and useful suggestions.

Specific Comments

1. Title. It would be good to include 'Central Atlantic Magmatic Province'. May be change this to 'Deep CO₂ in the end-Triassic Large Central Atlantic Magmatic province'?

The title has been modified to “Deep CO₂ in the end-Triassic Central Atlantic Magmatic Province”.

2. Line 42. There needs to be some mention that the study focuses on flood basalts, which volumetrically dominate Large Igneous Provinces, rather than the spatially & temporally associated alkaline igneous rocks (carbonatites etc).

This has been now mentioned at lines 40-41.

3. Line 51. A mention to the work of Karlstrom & Richards (2011) would seem appropriate here (and elsewhere in the manuscript). This was the first paper to quantitatively discuss the role of CO₂ during high-pressure fractionation of clinopyroxene in lower crustal magma chambers believed to be associated with many flood basalt plumbing systems.

We have now cited this paper at line 53.

4. Line 69. Issues related to gas exsolution bubbles within melt inclusions in flood basalts were discussed by Black & Gibson ((2019), pg 320). I appreciate this paper may have been published very shortly before the authors submitted their manuscript.

We have now cited this paper at lines 74 and 206.

5. Line 78. Change 'slightly' to 'minor'.

Changed as suggested.

6. Line 94. Change 'suggest' to 'suggests'

Changed as suggested.

7. Line 153. Please supply the analyses of the host clinopyroxenes and plagioclase feldspars. I'm assuming that the clinopyroxenes are sub-calcic augites as proposed to crystallise at high pressures in the experimental study of flood basalts by Thompson (1974)? Regardless, the reader is not able to calculate the crystallisation depths of the clinopyroxenes (using the method of Putirka etc) without this information. To do this, the composition of the clinopyroxene that hosts each melt inclusion needs to be clearly defined in a table. This information is required to calculate pressure and vital to calculating the CO₂ contents of the melt inclusions (Line 153).

Good point. In the *Supplementary Information* a table (Supplementary Tab. 4), containing the compositions of host clinopyroxene used for geothermobarometry estimates (Supplementary Fig. 4), has been added. We didn't add any measured composition of plagioclase or other mineral phases, since they were not used for geothermobarometry estimates. In the specific case of plagioclase, the reference to Marzoli et al. (2019), along with the label of the analysis, has been provided for plagioclase composition

in the *Supplementary Information*, in the section titled *Chemical composition of the MIs glass and its origin*.

8. Line 159. Please add information to the main text related to how the total estimate of 10^5 Gt for the Central Atlantic Magmatic Province was made. This needs to include some of the following details that were provided post submission:

‘Starting from the minimum calculated values of the CO_2 concentration in our melt inclusions (i.e., 0.5-0.6 wt. %) as representative of CAMP magma, we evaluated the total amount of degassed volcanic CO_2 during CAMP emplacement (i.e., up to 10^5 Gt). For this calculation, we considered an average density of 2.90 g/cm^3 for basaltic rocks and $5-6 \times 10^6 \text{ km}^3$ for the total volume of CAMP, in order to take into account also the deep plumbing system (“Over $2 \times 10^6 \text{ km}^3$ of CAMP magmas” is reported at line 60 of the Main Text)’

We have now added these details in the *Main Text* at lines 199-203.

9. The total estimate of 10^5 Gt erupted over 400,000 yrs is similar to the 3000 Mt per year estimated by Black & Gibson (2019) and merits some discussion.

We have now discussed this point along with the following point of the reviewer, at lines 204-206.

10. Lines 156 to 160. The concentrations of CO_2 (0.5 to 1 wt. %) that the authors calculate for the melts trapped in clinopyroxene from the CAMP are almost identical to (i) those estimated for the North Atlantic Province and Siberian Traps and (ii) CO_2 estimates predicted for the amounts of melting associated with the generation of tholeiitic magmas in upwelling mantle plumes (see Black & Gibson, 2019). These studies involved different approaches (volatiles versus trace element ratios of melt inclusions, numerical melt models) but it is surely noteworthy that the findings from the study by Capriolo et al are consistent with those estimated for flood basalts erupted in different Large Igneous Provinces.

Useful point that we briefly discussed at lines 204-206.

In summary, this is a well-written and nicely illustrated manuscript. The work is innovative, the data high quality and the interpretations are exciting. The paper will be of great interest to the readers of Nature Communications and, subject to the minor points above being addressed, I believe that it is worthy of publication.

Sally A Gibson

21st November 2019

References

Black, B.A., Gibson, S.A., 2019. Deep Carbon and the Life Cycle of Large Igneous Provinces. *Elements* 15, 319–324.

Karlstrom, L., Richards, M., 2011. On the evolution of large ultramafic magma chambers and timescales for flood basalt eruptions. *J. Geophys. Res.* 116, 13 PP. <https://doi.org/201110.1029/2010JB008159>

Thompson, R.N., 1974. Primary basalts and magma genesis. *Contrib. Mineral. Petrol.* 45, 317–341.

Reviewer #2 (Remarks to the Author):

Deep CO₂ in the end-Triassic Large Igneous Province

I am happy to provide an opinion as originally requested on the significance of the advance of the authors study and the links to the end-Triassic extinction event.

This paper uses new observations on the contents of gas exsolution bubbles contained in melt inclusions within mainly pyroxene crystals, in order to argue for large quantities of carbon-dioxide from deep crustal and subcrustal sources in large igneous province melts. The evidence is very clearly set out and coherently argued and, if correct, represents an important contribution to our understanding of the relationship between large igneous province formation and mass extinctions and/or extreme climate change. This paper is particularly significant because it focusses one of the largest but least well understood mass extinctions in the Phanerozoic, in the latest Triassic. The real advance here would be viability of deep carbon reservoirs (and presumably other deeply sourced volatiles) as the principal driver of environmental change, rather than the shallow sources which are currently the main focus of research.

This is a multidisciplinary study and I do not feel well qualified to provide comment on the deep crustal or mantle processes discussed, nor on the details of the analytical methods – it is important that these aspects of the paper are separately reviewed. However, the basic observations do seem to me to be strong, and also based on analysis of material from a sufficiently wide range of samples from different parts of the the Central Atlantic Magmatic Province in order to allow the broader conclusions to be reached.

My main criticism of the paper as written is the balance between material in the main text and the supplement. The results section (lines 75–104) makes repeated reference to Supplementary Information (tables and figures), which I personally found frustrating. These results are the bedrock on which the interpretations are based, and should be a central part of the main text. On the other hand, figures in the main text (Figure 3) include examples of the Raman spectra which, although also important, is the kind of more detailed evidence that may be best placed in supplementary data. Additionally, Figs 1 and 2 contain multiple examples of essentially the same phenomena. I recommend, then, that the balance of material in the main text and the supplementary information is reconsidered, such that the main observational results are fully illustrated in the main text.

We thank the reviewer for his time and are encouraged by his positive assessment. We agree that the balance between *Main Text* and *Supplementary Information* could be improved and have now added most of the sections “*Results*” and “*CO₂ in CAMP basalts*” to the *Main Text* (previously present in the *Supplementary Information* only). The manuscript now features additional figures in the *Main Text* (sample location, SEM-EDS maps, and a crossplot of the Raman spectra of elemental carbon), and includes broader discussions. Overall we feel this has improved the paper. Given the nature of these changes we have also been over the whole text to ensure coherency throughout and made minor and grammatical changes.

Reviewers' comments (received on 27th January 2020)

Reviewer #1 (Remarks to the Author):

The authors have made all of the modifications that I suggested and I believe that the manuscript is now acceptable for publication. [Please check spelling of sulphide vs sulphide]